# Eliciting Thinking Hierarchy without a Prior

**Yuqing Kong**[*]
CFCS and School of Computer Science
Peking University
yuqing.kong@pku.edu.cn

**Yunqi Li**[†]
CFCS and School of EECS
Peking University
liyunqi@pku.edu.cn

**Yubo Zhang**
CFCS and School of Computer Science
Peking University
falsyta@pku.edu.cn

**Zhihuan Huang**
CFCS and School of Computer Science
Peking University
zhihuan.huang@pku.edu.cn

**Jinzhao Wu**[‡]
CFCS and School of EECS
Peking University
jinzhao.wu@pku.edu.cn

## Abstract

When we use the wisdom of the crowds, we usually rank the answers according to their popularity, especially when we cannot verify the answers. However, this can be very dangerous when the majority make systematic mistakes. A fundamental question arises: can we build a hierarchy among the answers *without any prior* where the higher-ranking answers, which may not be supported by the majority, are from more sophisticated people? To address the question, we propose 1) a novel model to describe people's thinking hierarchy; 2) two algorithms to learn the thinking hierarchy without any prior; 3) a novel open-response based crowd-sourcing approach based on the above theoretic framework. In addition to theoretic justifications, we conduct four empirical crowdsourcing studies and show that a) the accuracy of the top-ranking answers learned by our approach is much higher than that of plurality voting (In one question, the plurality answer is supported by 74 respondents but the correct answer is only supported by 3 respondents. Our approach ranks the correct answer the highest without any prior); b) our model has a high goodness-of-fit, especially for the questions where our top-ranking answer is correct. To the best of our knowledge, we are the first to propose a thinking hierarchy model with empirical validations in the general problem-solving scenarios; and the first to propose a practical open-response based crowdsourcing approach that beats plurality voting without any prior.

## 1 Introduction

The wisdom of the crowds has been proved to lead to better decision-making and problem-solving than that of an individual, especially when we do not have sufficient prior knowledge to identify individual experts [26, 2, 28]. Plurality is one of the most popular ways to aggregate the crowd's opinions. The opinions are usually ranked according to their popularity. However, it can be very

---

[*]Corresponding Author

[†]This author is currently at Stanford University

[‡]This author is currently at Yale University

36th Conference on Neural Information Processing Systems (NeurIPS 2022).

dangerous when the majority are systematically biased. Here is a real-world study we perform. We have asked multiple top university students the following question.

*The radius of Circle A is 1/3 the radius of Circle B. Circle A rolls around Circle B one trip back to its starting point. How many times will Circle A revolve in total?*

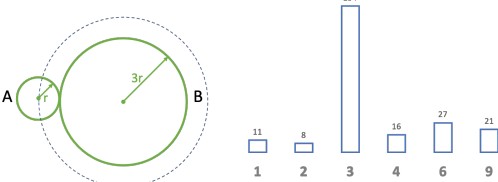

Figure 1: Collected answers of the circle problem

We have collected answers "1 (11 people), 2 (8 people), 3 (134 people), 4 (16 people), 6 (27 people), 9 (21 people)". The plurality answer is "3", the ratio between the big circle's circumference and the small circle's. However, the correct answer is "4"[4] which is only supported by 16 people.

With prior knowledge like the expertise level of each individual respondent, we may be able to identify the correct answer. However, sometimes it's quite expensive and difficult to obtain prior knowledge, especially in new fields.

To address the above issue, Prelec et al. [19] propose an innovative approach, surprisingly popular. They prepare multiple choices, ask the respondents to pick one option, and more importantly, predict the distribution over other people's choices. They use the predictions to construct a prior distribution over the choices, and then select the choice that is more popular than the prior such that the bias is corrected. Many other work [11, 6, 20] develop the idea of using the prior or the predictions to correct the bias.

However, first, it's not applicable to employ the previous approaches into the running example, the circle problem, because they require prior knowledge to design the choices. It's also effortful for respondents to report a whole distribution over all choices.

Second, previous works focus on using the predictions to correct bias, while it's intrinsically interesting to build a thinking hierarchy using their predictions. This leads to a hierarchy among the answers as well. The famous cognitive hierarchy theory (CHT) [24, 25, 3] builds a thinking theory in the scenarios when people play games such that we can learn the actions of players of different sophistication levels. Nevertheless, CHT is designed only for a game-theoretic setting.

We are curious about building a thinking theory in general problem-solving scenarios. The key insight is that people of a more sophisticated level know the mind of lower levels, but not vice versa [3, 10]. We want to apply the insight to learn the answers of people of different sophistication levels, called the *thinking hierarchy*, without any prior.

**Key question** We aim to build a practical approach to learn the thinking hierarchy *without any prior*. Based on the thinking hierarchy, we can rank the answers such that the higher-ranking answers, which may not be supported by the majority, are from more sophisticated people.

In addition to building a thinking theory in the general problem-solving scenarios, in practice, there are multiple reasons why we want the hierarchy rather than only the best. First, for some questions like subjective questions (e.g. why are bar chairs high?), there may be more than one high-quality answer and the full hierarchy provides a richer result. Second, the hierarchy among the answers helps to understand how people think better, which is important especially when we try to elicit people's opinions about a policy.

**Our approach** We follow the framework of asking for answers and predictions simultaneously and extend it to a more practical open-response based paradigm. The paradigm asks a single open

---

[4]Interested readers are referred to `https://math.stackexchange.com/questions/1351058/circle-revolutions-rolling-around-another-circle` for explanations.

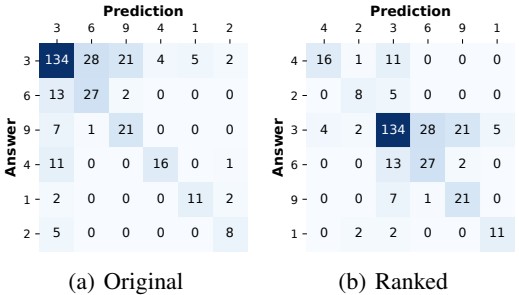

(a) Original       (b) Ranked

Figure 2: The empirical results of the circle problem.

response question and asks for both each respondent's answer and prediction [5] for other people's answers. For example, in the circle problem, a respondent can provide: answer: "4", prediction: "3". We then construct an answer-prediction matrix $\mathbf{A}$ that records the number of people who report a specific answer-prediction pair (e.g. Figure 2(a) shows that 28 people answer "3" and predict other people answer "6".).

To learn the thinking hierarchy, we propose a novel model. Our model describes how people of different sophistication levels answer the question and more importantly, predict other people's answers. The joint distribution over a respondent's answer and prediction depends on the latent parameters that describe people's thinking hierarchy. We show that given the joint distribution over a respondent's answer and prediction, we can infer the latent thinking hierarchy by solving a new variant of the non-negative matrix factorization problem, called non-negative congruence triangularization (NCT), which may be of an independent interest. Based on the analysis of NCT, we provide two simple answer-ranking algorithms and show that with proper assumptions, the algorithms will learn the latent thinking hierarchy given the joint distribution over a respondent's answer and prediction.

Finally, we show that the answer-prediction matrix collected by the paradigm is a proxy for the joint distribution over a respondent's answer and prediction. We implement the NCT based answer-ranking algorithms on the answer-prediction matrix. The default algorithm ranks the answers to maximize the sum of the square of the elements in the upper triangular area of the matrix. In a variant version, to allow different answers to have the same sophistication level, the answers are partitioned to compress the matrix. The algorithm maximizes the sum of the square of the upper triangular area of the compressed matrix.

In addition to the above theoretic framework, we also run empirical studies by asking people questions in various areas including math, Go, general knowledge, and character pronunciation.

**Example 1.1** (Empirical results of the circle problem). *In the circle problem, we have collected the empirical answer-prediction matrix (Figure 2(a)) and ranked it (Figure 2(b)) based on the default algorithm. The default ranking algorithm does not use any diagonal element. Thus, for ease of illustration, the diagonal elements are modified (i.e., for all $a$, $A_{a,a}$ is modified to the number of respondents who answer $a$) such that we can compare our method to the plurality voting visually.*

More empirical results will be illustrated in Section 3.1. We show the superiority of our algorithm by comparing our algorithm to plurality voting by the accuracy of the top-ranking answers. We also test the goodness-of-fit of our model based on the collected data set. To summarize, we provide a novel theoretic framework to study people's thinking hierarchy in the problem-solving scenarios and a practical open-response based crowdsourcing approach that outputs a high-quality answer only supported by 16 people when the wrong answer is supported by 134 people (another question is 3 vs. 74) without any prior.

---

[5]Unlike previous work, the prediction in our model is not a distribution but an answer the respondent thinks other people may answer.

## 1.1 Related work

**Information aggregation with the second order information**    Prelec [18], Prelec et al. [19] start the framework that asks the respondents both their answers and predictions for the distribution over other people's answers. However, to implement the framework, the requester needs to design a multi-choice question whose choices may require prior knowledge. The effort for a distribution report is non-minimal to many respondents and the quality can be an issue since most people are not perfect Bayesian. Kong and Schoenebeck [10] study how to elicit thinking hierarchy theoretically by people's predictions and also assumes that more sophisticated people can reason about the mind of less sophisticated people. However, they focus on multi-choice questions, and in their framework, agents either need to perform multiple tasks or report non-minimal distributions. Thus, it's difficult to perform empirical studies using their framework and there is also no empirical validation. Moreover, they assume more sophisticated people can reason about the mind of ALL less sophisticated people while our model allows more sophisticated people cannot reason about some less sophisticated people. Two recent works Hosseini et al. [7], Schoenebeck and Tao [21] study how to aggregate people's votes to rank a set of predetermined candidates (e.g. ranking paintings based on price [7], or two presidential candidates [21]) better by using people's predictions. Both of them treat the pairwise comparisons of the candidates as the elicited signals and use people's predictions to improve the signal quality. In contrast, rather than eliciting ranking based on price or preference, we elicit the thinking hierarchy among people by using people's predictions for other people to determine the hierarchy over the answers. Like Prelec et al. [19], Hosseini et al. [7], Schoenebeck and Tao [21], there are other works that use people's predictions to reduce the bias of the collected feedback. For example, Dasgupta et al. [6] use people's predictions to reduce the bias caused by interactions between users on a social network. Rothschild and Wolfers [20] show that voters' expectations for other people's votes are more informative than their intentions. In addition to the discrete setting, recently, a growing literature, including Palley and Soll [16], Martinie et al. [12], Chen et al. [4], Wilkening et al. [29], Palley and Satopää [15], Peker [17], aggregate forecasts with additional second order information like each forecaster's expectation for the average of other forecasters' forecasts.

**Peer prediction**    Starting from Miller et al. [13], a series of work (e.g.[13, 18, 5, 22, 11, 9]) focus on eliciting information without a prior by designing incentive-compatible mechanisms. This field is called peer prediction, or information elicitation without prior. In contrast, this work focus on how to aggregate information and identify high-quality information without a prior. Liu et al. [11], Prelec et al. [19], Hosseini et al. [7], Schoenebeck and Tao [21] focus on information aggregation without prior. However, they do not focus on the problem of learning thinking hierarchy like this work.

**Bounded rationality**    Starting from Simon [23], the term "bounded rationality" describes the decision maker's cognitive limitations. Stahl [24] propose a behavioral model of bounded rationality to predict people's behaviors in strategic games, the level-k theory. Level-k theory assumes that players have different levels of sophistication. Level-0 players play non-strategically, level-1 players play optimal response to level-0 players... A variant of level-k theory is cognitive hierarchy theory [25, 3] where level-k players believe that lower-level players' percentages follow a certain type of distribution. Our Thinking hierarchy model is conceptually similar to level-k but focuses on a non-game setting. Moreover, the level-k theory focus on estimating the average levels of a population while we focus on identifying each respondent's level.

## 2   Learning Thinking Hierarchy

We first introduce our model for thinking hierarchy. Tversky and Kahneman [27] propose that people have two systems, a fast and intuitive system, and a slow and logical system. For example, Alice starts to solve the circle problem. When she reads the question, she can run her intuitive system 1 and obtain answer "3". However, when she starts to think carefully, she runs her more careful system 2 to obtain answer "4".

We propose a more general model where people can run multiple oracles to approach the question during their thinking process. Conceptually similar to the cognitive hierarchy theory [3], we assume the oracles have different levels. People usually run lower level oracles before the higher level oracles. In the previous example, system 1 is the lower level oracle and system 2 is the higher one. We assume that Alice runs system 2 after system 1. Thus, in our model, people who know the answer is "4" can

predict that other people answer "3", which is hypothesized in Kong and Schoenebeck [10]. It's also possible that some people run the most sophisticated oracle directly without running the lower ones. Thus, we design the model such that it does not require the higher-type to be able to predict ALL lower types. We also allow the oracles' outputs to be random.

Section 2.1 introduces the model of thinking hierarchy. Section 2.2 reduces the model inference problem to a matrix decomposition problem. Section 2.3 and section 2.4 show how to learn the thinking hierarchy.

## 2.1 Thinking hierarchy

Fixing a question $q$ (e.g. the circle problem), $T$ denotes the set of thinking types. $A$ denotes the set of possible answers. Both $T$ and $A$ are finite sets. $\Delta_A$ denotes all possible distributions over $A$. We sometimes use "prob" as a shorthand for probability.

**Generating answers**   We will describe how people of different thinking types generate answers.

**Definition 2.1** (Oracles of thinking types $\mathbf{W}$ ). *An answer generating oracle maps the question to an (random) answer in A. Each type $t$ corresponds to an oracle $O_t$. The output $O_t(q)$ is a random variable whose distribution is $\mathbf{w}_t \in \Delta_A$. $\mathbf{W}$ denotes a $|T| \times |A|$ matrix where each row $t$ is $\mathbf{w}_t$.*

Each respondent is type $t$ with prob $p_t$ and $\sum_t p_t = 1$. A type $t$ respondent generates the answer by running the oracle $O_t$. For all $a \in A$, the probability that a respondent answers $a$ will be $\sum_t p_t \mathbf{w}_t(a)$. We assume the probability is positive for all $a \in A$.

**Example 2.2** (A running example). *There are two types $T = \{0, 1\}$. The answer space is $A = \{3, 4, 6\}$. $O_0$ will output '3' with probability 0.8 and '6' with probability 0.2. $O_1$ will output '4' deterministically. In this example, $\mathbf{W} = \begin{bmatrix} 0.8 & 0 & 0.2 \\ 0 & 1 & 0 \end{bmatrix}$ where the first row is the distribution of $O_0$'s output and the second row is the distribution of $O_1$'s output.*

**Generating predictions**   We then describe how people of different thinking types predict what other people will answer. Here the prediction is not a distribution, but an answer other people may report. When a type $t$ respondent makes a prediction, she will run an oracle, which is $O_{t'}$ with probability $p_{t \to t'}$ where $\sum_{t'} p_{t \to t'} = 1$. She uses the output of $O_{t'}$ as the prediction $g \in A$.

**Combination: answer-prediction joint distribution $\mathbf{M}$**   $\mathbf{M}$ denotes a $|A| \times |A|$ matrix where $M_{a,g}$ is the probability an respondent answers $a$ and predicts $g$. $\mathbf{\Lambda}$ denotes a $|T| \times |T|$ matrix where $\Lambda_{t,t'} = p_t p_{t \to t'}$ is the probability a respondent is type $t$ and predicts type $t'$.

**Example 2.3.** *In this example, when type 0 respondent makes a prediction, with prob 1, she will run $O_0$ again. When type 1 respondent makes a prediction, with prob 0.5, she will run $O_1$ again, with prob 0.5, she will run $O_0$. Moreover, a respondent is type 0 with prob 0.7, and type 1 with prob 0.3. Here $\mathbf{\Lambda} = \begin{bmatrix} p_0 p_{0 \to 0} & p_0 p_{0 \to 1} \\ p_1 p_{1 \to 0} & p_1 p_{1 \to 1} \end{bmatrix} = \begin{bmatrix} 0.7*1 & 0.7*0 \\ 0.3*0.5 & 0.3*0.5 \end{bmatrix} = \begin{bmatrix} 0.7 & 0 \\ 0.15 & 0.15 \end{bmatrix}$.*

**Claim 2.4.** *Based on the above generating processes, $\mathbf{M} = \mathbf{W}^\top \mathbf{\Lambda} \mathbf{W}$.*

*Proof.* For each respondent, the probability she answers $a$ and predicts $g$ will be

$$M_{a,g} = \sum_t p_t \mathbf{w}_t(a) \sum_{t'} p_{t \to t'} \mathbf{w}_{t'}(g) = \sum_{t,t'} \mathbf{w}_t(a) p_t p_{t \to t'} \mathbf{w}_{t'}(g).$$

We sum over all possible types the respondent will be. Given she is type $t$, she runs oracle $O_t$ to generate the answer and $\mathbf{w}_t(a)$ is the probability that the answer is $a$. We sum over all possible oracles she runs to predict. Given that she runs $O_{t'}$, $\mathbf{w}_{t'}(g)$ is the probability the prediction is $g$. □

**Key assumption: upper-triangular $\mathbf{\Lambda}$**   we assume that people of a less sophisticated level can never run the oracles of more sophisticated levels. A linear ordering of types $\pi : \{1, 2, \cdots, |T|\} \mapsto T$ maps a ranking position to a type. For example, $\pi(1) \in T$ is the top-ranking type.

**Assumption 2.5.** *We assume that with a proper ordering $\pi$ of the types, $\mathbf{\Lambda}$ is an upper-triangular matrix. Formally, there exists $\pi$ such that $\forall i > j, \mathbf{\Lambda}_{\pi(i),\pi(j)} = 0$. Any $\pi$ that makes $\mathbf{\Lambda}$ upper-triangular is a valid thinking hierarchy of the types.*

In the running example, the valid thinking hierarchy is $\pi(1) = $ type 1, $\pi(2) = $ type 0. Note that the above assumption does not require $\forall i \leq j, \mathbf{\Lambda}_{\pi(i),\pi(j)} > 0$. When $\mathbf{\Lambda}$ is a diagonal matrix, types cannot predict each other and are equally sophisticated, thus any ordering is a valid thinking hierarchy.

An algorithm **finds the thinking hierarchy** when the algorithm is given $\mathbf{M}$ which is generated by latent (unknown) $\mathbf{W}, \mathbf{\Lambda}$, and the algorithm will output a matrix $\mathbf{W}^*$ which equals a row-permuted $\mathbf{W}$ where the row order is a valid thinking hierarchy. Formally, there exists a valid thinking hierarchy $\pi$ such that the $i^{th}$ row of $\mathbf{W}^*$ is the $\pi(i)^{th}$ row of $\mathbf{W}$, i.e, $\mathbf{w}_i^* = \mathbf{w}_{\pi(i)}$.

## 2.2 Non-negative Congruence Triangularization (NCT)

With the above model, inferring thinking hierarchy leads to a novel matrix decomposition problem, which is similar to the symmetric non-negative matrix factorization problem (NMF)[6]. A non-negative matrix is a matrix whose elements are non-negative.

**Definition 2.6** (Non-negative Congruence[7] Triangularization (NCT)). *Given a non-negative matrix* $\mathbf{M}$, *NCT aims to find non-negative matrices* $\mathbf{W}$ *and non-negative upper-triangular matrix* $\mathbf{\Lambda}$ *such that* $\mathbf{M} = \mathbf{W}^\top \mathbf{\Lambda} \mathbf{W}$. *In a Frobenius norm based approximated version, given a set of matrices* $\mathcal{W}$, *NCT aims to find non-negative matrices* $\mathbf{W}$ *and non-negative upper-triangular matrix* $\mathbf{\Lambda}$ *to minimize*

$$\min_{\mathbf{W} \in \mathcal{W}, \mathbf{\Lambda}} ||\mathbf{M} - \mathbf{W}^\top \mathbf{\Lambda} \mathbf{W}||_F^2$$

*and the minimum is defined as the lack-of-fit of* $\mathbf{M}$ *regarding* $\mathcal{W}$[8].

Like NMF, it's impossible to ask for the strict uniqueness of the results. Let $P_{\mathbf{\Lambda}}$ be the set of permutation matrices such that $\mathbf{\Pi}^\top \mathbf{\Lambda} \mathbf{\Pi}$ is still upper-triangular. If $(\mathbf{W}, \mathbf{\Lambda})$ is a solution, then $(\mathbf{\Pi}^{-1}\mathbf{D}\mathbf{W}, \mathbf{\Pi}^\top \mathbf{D}^{-1}\mathbf{\Lambda}\mathbf{D}^{-1}\mathbf{\Pi})$ is also a solution where $\mathbf{D}$ is a diagonal matrix with positive elements and $\mathbf{\Pi} \in P_{\mathbf{\Lambda}}$. We state the uniqueness results as follows and the proof is deferred to Appendix C.

**Proposition 2.7** (Uniqueness). *If* $|T| \leq |A|$ *and* $T$ *columns of* $\mathbf{W}$ *consist of a permuted positive diagonal matrix, NCT for* $\mathbf{M} = \mathbf{W}^\top \mathbf{\Lambda} \mathbf{W}$ *is unique in the sense that then for all* $\mathbf{W}'^\top \mathbf{\Lambda}' \mathbf{W}' = \mathbf{W}^\top \mathbf{\Lambda} \mathbf{W}$, *there exists a positive diagonal matrix* $\mathbf{D}$ *and a* $|T| \times |T|$ *permutation matrix* $\mathbf{\Pi} \in P_{\mathbf{\Lambda}}$ *such that* $\mathbf{W}' = \mathbf{\Pi}^{-1}\mathbf{D}\mathbf{W}$.

When we restrict $\mathbf{W}$ to be "**semi-orthogonal**", we obtain a clean format of NCT without searching for optimal $\mathbf{\Lambda}$. $\mathcal{I}$ is the set of all "semi-orthogonal" matrices $\mathbf{W}$ where each column of $\mathbf{W}$ has and only has one non-zero element and $\mathbf{W}\mathbf{W}^\top = \mathbf{I}$. For example, the $\mathbf{W}$ in Example 2.2 can be normalized to a semi-orthogonal matrix. The following lemma follows from the expansion of the Frobenius norm and we defer the proof to Appendix C.

**Lemma 2.8** (Semi-orthogonal: minimizing F-norm = maximizing upper-triangular's sum of the square). *For all set of matrices* $\mathcal{W} \subset \mathcal{I}$, $\min_{\mathbf{W} \in \mathcal{W}, \mathbf{\Lambda}} ||\mathbf{M} - \mathbf{W}^\top \mathbf{\Lambda} \mathbf{W}||_F^2$ *is equivalent to solving* $\max_{\mathbf{W} \in \mathcal{W}} \sum_{i \leq j} (\mathbf{W}\mathbf{M}\mathbf{W}^\top)_{i,j}^2$ *and setting* $\mathbf{\Lambda}$ *as* $\mathrm{Up}(\mathbf{W}\mathbf{M}\mathbf{W}^\top)$, *the upper-triangular area of* $\mathbf{W}\mathbf{M}\mathbf{W}^\top$.

## 2.3 Inferring the thinking hierarchy with answer-prediction joint distribution $\mathbf{M}$

Given $\mathbf{M}$, inferring the thinking hierarchy is equivalent to solving NCT in general. Though we do not have $\mathbf{M}$, later we will show a proxy for $\mathbf{M}$. For simplicity of practical use, we introduce two simple ranking algorithms by employing Lemma 2.8. The ranking algorithm takes $\mathbf{M}$ as input and outputs a linear ordering of answers $\pi : \{1, 2, \cdots, |A|\} \mapsto A$ that maps a ranking position to an answer.

**Answer-Ranking Algorithm (Default)** $AR(\mathbf{M})$    The algorithm computes

$$\mathbf{\Pi}^* \leftarrow \arg\max_{\mathbf{\Pi} \in \mathcal{P}} \sum_{i \leq j} (\mathbf{\Pi}\mathbf{M}\mathbf{\Pi}^\top)_{i,j}^2$$

---

[6]Symmetric NMF: $\min_{\mathbf{W}} ||\mathbf{M} - \mathbf{W}^\top \mathbf{W}||_F^2$

[7]We use congruence here though it is not matrix congruence since $\mathbf{W}$ may not be a square matrix.

[8]$\mathbf{M} = \mathbf{W}^\top \mathbf{\Lambda} \mathbf{W}, \mathbf{W} \in \mathcal{W}$ has zero lack-of-fit.

where $\mathcal{P}$ is the set of all $|A| \times |A|$ permutation matrices. There is a one to one mapping between each permutation matrix $\mathbf{\Pi}$ and a linear ordering $\pi$: $\Pi_{i,\pi(i)} = 1, \forall i$. Therefore, the optimal $\mathbf{\Pi}^*$ leads to an optimal rank over answers directly and the default algorithm can be also represented as

$$\pi^* \leftarrow \arg\max_\pi \sum_{i \leq j} M^2_{\pi(i),\pi(j)}.$$

To find the optimal rank, we use a dynamic programming based algorithm (see the supplementary materials) which takes $O(2^{|A|}|A|^2)$. In practice, $|A|$ is usually at most 7 or 8. In our empirical study, the default algorithm takes 91 milliseconds to finish the computation of all 152 questions.

The default algorithm implicitly assumes $|T| = |A|$ and all oracles are deterministic. To allow $|T| < |A|$ and non-deterministic oracles, we introduce a variant that generalizes $\mathcal{P}$ to a subset of semi-orthogonal matrices $\mathcal{I}$. Every $|T| \times |A|$ semi-orthogonal $\mathbf{W}$ indicates a hard clustering. Each cluster $t \in T$ contains all answers $a$ such that $W_{t,a} > 0$. For example, the $\mathbf{W}$ in Example 2.2 can be normalized to a semi-orthogonal matrix and indicates a hard clustering $\{4\}, \{6, 3\}$. Therefore, the variant algorithm will partition the answers into multiple clusters and assign a hierarchy to the clusters.

**Answer-Ranking Algorithm (Variant)** $\quad AR^+(\mathbf{M}, \mathcal{W})$ The algorithm computes

$$\mathbf{W}^* \leftarrow \arg\max_{\mathbf{W} \in \mathcal{I}} \sum_{i \leq j} (\mathbf{W}\mathbf{M}\mathbf{W}^\top)^2_{i,j}.$$

where $\mathcal{W} \subset \mathcal{I}$. $\mathbf{W}^*$ is normalized such that every row sums to 1. This algorithm does not restrict $|T| = |A|$ and learns the optimal $|T|$.

**$\mathbf{W}^* \Rightarrow$ Answer rank** The output $\mathbf{W}^*$ indicates a hard clustering of all answers. We rank all answer as follows: for any $i < j$, the answers in cluster $i$ has a higher rank [9] than the answers in cluster $j$. For all $i$, for any two answers $a, a'$ in the same cluster $i$, $a$ is ranked higher than $a'$ if $W^*_{i,a} > W^*_{i,a'}$.

**Theoretical justification** When $\mathbf{M}$ perfectly fits the model with the restriction that the latent $\mathbf{W}$ is a permutation or a hard clustering, we show that our algorithm finds the thinking hierarchy. Otherwise, our algorithm finds the "closest" solution measured by the Frobenius norm.

**Theorem 2.9.** *When there exists $\mathbf{\Pi}_0 \in \mathcal{P}$ and non-negative upper-triangular matrix $\mathbf{\Lambda}_0$ such that $\mathbf{M} = \mathbf{\Pi}_0^\top \mathbf{\Lambda}_0 \mathbf{\Pi}_0$, $AR(\mathbf{M})$ finds the thinking hierarchy [10]. In general, $AR(\mathbf{M})$ will output $\mathbf{\Pi}^*$ where $\mathbf{\Pi}^*, \mathbf{\Lambda}^* = \mathrm{Up}(\mathbf{\Pi}^*\mathbf{M}\mathbf{\Pi}^{*\top})$ is a solution to $\arg\min_{\mathbf{\Pi} \in \mathcal{P}, \mathbf{\Lambda}} ||\mathbf{M} - \mathbf{\Pi}^\top \mathbf{\Lambda}\mathbf{\Pi}||_F^2$. The above statement is still valid by replacing $\mathcal{P}$ by $\mathcal{W} \subset \mathcal{I}$ and $AR(\mathbf{M})$ by $AR^+(\mathbf{M}, \mathcal{W})$.*

Proposition 2.7 and Lemma 2.8 almost directly imply the above theorem. We defer the formal proof to Appendix C.

## 2.4 A proxy for answer-prediction joint distribution $\mathbf{M}$

In practice, we do not have perfect $\mathbf{M}$. We use the following *open-response* paradigm to obtain a proxy for $\mathbf{M}$.

**Answer-prediction paradigm** the respondents are asked *Q1: What's your answer? Answer:_____ Q2: What do you think other people will answer:_____.*

In the circle example, the possible feedback can be "answer: 4; prediction: 3", "answer: 3; prediction: 6,9,1"... We collect all answers provided by the respondents [11] and denote the set of them by $A$. In the circle example, $A = \{1, 2, 3, 4, 6, 9\}$. We also allow respondents to provide no prediction or multiple predictions.

---

[9]Rank 1 answer is in the highest position.

[10]See definition in the last paragraph in Section 2.1.

[11]In practice, we set a threshold $\theta$ and collect answers which are provided by at least $\theta$ fraction of the respondents. We allow multiple predictions and also allow people to answer "I do not know". See Section 3 for more details.

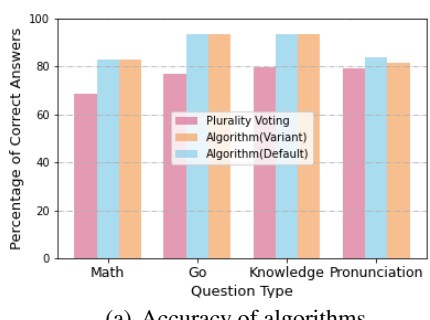
(a) Accuracy of algorithms

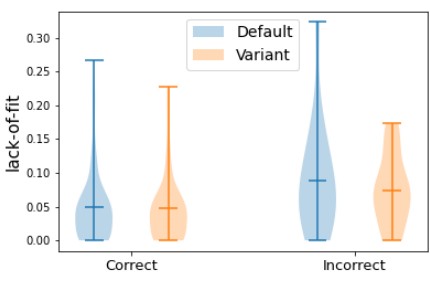
(b) Empirical distribution of lack-of-fit

Figure 3: The results of our experiment

**Answer-prediction matrix**   We aggregate the feedback and visualize it by an Answer-Prediction matrix. The Answer-Prediction matrix $\mathbf{A}$ is a $|A| \times |A|$ square matrix where $|A|$ is the number of distinct answers provided by the respondents. Each entry $A_{a,g}, a, g \in A$ is the number of respondents that answer "a" and predict "g".

We will show that with proper assumptions, the answer-prediction matrix $\mathbf{A}$'s expectation is proportional to $\mathbf{M}$. First, for ease of analysis, we assume that each respondent's predictions are i.i.d. samples[12]. Second, since we allow people to optionally provide predictions, we need to additionally assume that the number of predictions each respondent is willing to provide is independent of her type and answer. We state the formal result as follows and the proof is deferred to Appendix C.

**Proposition 2.10.** *When each respondent's predictions are i.i.d. samples, and the number of predictions she gives is independent of her type and answer, the answer-prediction matrix $\mathbf{A}$'s expectation is proportional to $\mathbf{M}$.*

## 3 Studies

We conduct four studies, study 1 (35 math problems), study 2 (30 Go problems), study 3 (44 general knowledge questions), and study 4 (43 Chinese character pronunciation questions).

**Data collection**   All studies are performed by online questionnaires. We recruit the respondents by an online announcement[13] or from an online platform that is similar to Amazon Mechanical Turk. We get respondents' consent for using and sharing their data for research. Respondents are asked not to search for the answers to the questions or communicate with other people. We allow the respondents to answer "I do not know" for all questions. Except for Go problems, all questionnaires use flat payment. We illustrate the data collection process in detail in Appendix A. We allow respondents to participate in more than one study because our algorithms analyze each question separately and independently.

**Data processing**   We merge answers which are the same, like '0.5' and '50%'. We omit the answers that are reported by less than ($\leq$) 3% of respondents or one person. The remaining answers, excluding "I do not know", form the answer set $A$ whose size is $|A|$. We then construct the Answer-Prediction matrix and perform our algorithms. Pseudo-codes are attached in Appendix B. Our algorithms do not require any prior or the respondents' expertise levels.

### 3.1 Results

---

[12]This may not be a very good assumption since i.i.d. samples can repeat but respondents usually do not repeat their predictions. If we do not want this assumption, we can choose to only use the first prediction from each respondent (if there exists) to construct the answer-prediction matrix.

[13]Many are students from top universities in China. See Appendix A for more details.

| Type | Total | Our algorithm(Default) | Our algorithm(Variant) | Plurality voting |
|---|---|---|---|---|
| Math | 35 | **29** | **29** | 24 |
| Go | 30 | **28** | **28** | 23 |
| General knowledge | 44 | **41** | **41** | 35 |
| Chinese character | 43 | **36** | 35 | 34 |

Table 1: The number of questions our algorithms/baseline are correct.

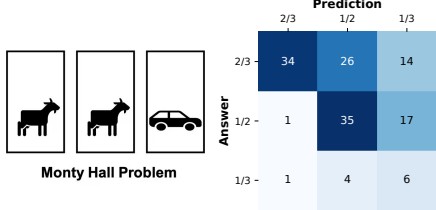

(a) the Monty Hall problem: you can select one closed door of three. A prize, a car, is behind one of the doors. The other two doors hide goats. After you have made your choice, Monty Hall will open one of the remaining doors and show that it does not contain the prize. He then asks you if you would like to switch your choice to the other unopened door. What is the probability to get the prize if you switch?

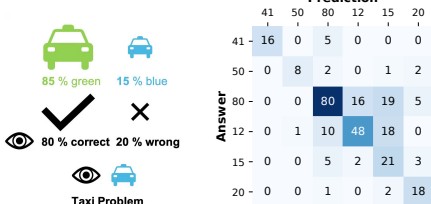

(b) the Taxicab problem: 85% of taxis in this city are green, the others are blue. A witness sees a blue taxi. She is usually correct with probability 80%. What is the probability that the taxi saw by the witness is blue?

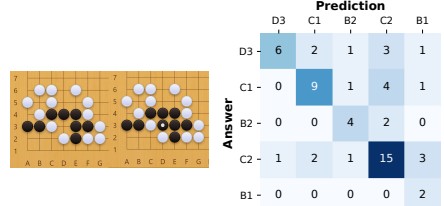

(c) Pick a move for black such that they can be alive.

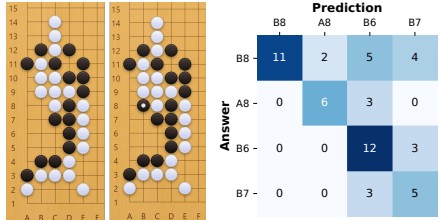

(d) Pick a move for black such that they can be alive by ko.

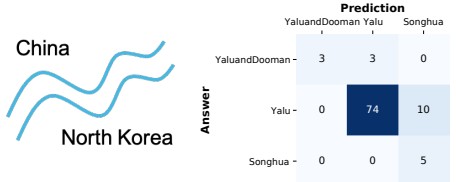

(e) the boundary question: what river forms the boundary between North Korea and China?

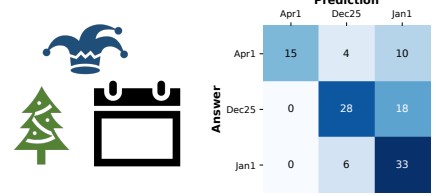

(f) the Middle Age New Year question: when was the new year in middle age?

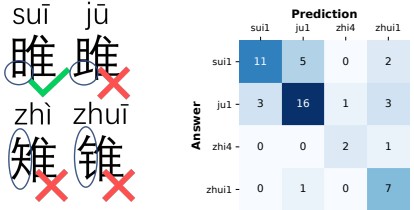

(g) the pronunciation of 睢

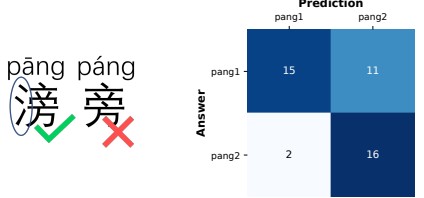

(h) the pronunciation of 滂

Figure 4: The ranked answer-prediction matrices

We compare our approach to the baseline, the plurality voting, regarding the accuracy of the top-ranking answers. Both of the algorithms beat plurality voting for all studies and the default is slightly better. Among all 152 questions, in 138 questions, the variant algorithm outputs the same hierarchy as the default algorithm. For other questions, the top-ranking type of the variant algorithm may contain more than one answer. The top-ranking answer is the answer supported by more people among all answers in the top-ranking type. In one question the variant is wrong but the default is correct, the variant algorithm assigns both the correct answer and the incorrect plurality answer to the top-ranking type, thus outputting the incorrect answer as the top-ranking answer.

We also compute the lack-of-fit index (Definition 2.6) of the algorithms and find that the questions the algorithm outputs the correct answer have a smaller lack-of-fit thus fitting the thinking hierarchy model better. Therefore, we can use the lack-of-fit index as a reliability index of the algorithms.

We additionally pick several representative examples for each study (Figure 4) where the matrices are ranked by the default algorithm and the diagonal area modified like Example 1.1. In all of these examples, the plurality is incorrect while our approach is correct. Results of other questions are illustrated at `https://elicitation.info/classroom/1/`. Detailed explanations are illustrated in Appendix D and here we provide some highlights. First, our approach elicits a rich hierarchy. For example, the taxicab problem is borrowed from Kahneman [8] and previous studies show that people usually ignore the base rate and report '80%'. The imagined levels can be 41%→80%. We elicit a much richer level "**41%**→50%→80%→12%→15%→20%". Second, the most sophisticated level may fail to predict the least one. In the Taxicab problem, the correct "41%" supporters successfully predict the common wrong answer "80%". However, they fail to predict the surprisingly wrong answers "12%,20%", which are in contrast successfully predicted by "80%" supporters. Our model is sufficiently general to allow this situation. Third, even for problems (like Go) without famous mistakes, our approach still works. Moreover, in the boundary question, we identify the correct answer without any prior when only 3 respondents are correct.

## 4   Discussion

One future direction is to consider incentives in our paradigm like the literature of information elicitation without verification [13, 18, 5, 22, 9]. We have asked a class of students at Peking University: *why are bar chairs high?* using our paradigm. We cluster the answers by hand. The plurality answer is "the bar counter is high" and our top-ranking answer is "better eye contact with people who stand". Thus, another future diction is to extend our approach to the scenario where people's answers are sentences, where we can apply NLP to cluster them automatically. In summary, we propose the first empirically validated method to learn the thinking hierarchy without any prior in the general problem-solving scenarios. Potentially, our paradigm can be used to make a better decision when we crowd-source opinions in a new field with little prior information. Moreover, when we elicit the crowds' opinions for a policy, with the thinking hierarchy information, it's possible to understand the crowds' opinions better. However, regarding the negative impact, it may be easier to implement a social media manipulation of public opinion with the full thinking hierarchy of the crowds. One interesting future direction is to explore the impact of the thinking hierarchy information.

## Acknowledgement

This research is supported by National Natural Science Foundation of China award number 62002001. We would like to thank Xiaotie Deng, Xiaoming Li, Grant Schoenebeck, and David Parkes for their useful suggestions, and all participants of our studies for their time and efforts.

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
