# OpenReview forum: "Eliciting Thinking Hierarchy without a Prior"
_NeurIPS.cc/2022/Conference — NeurIPS 2022 Accept_

### Official Review · Reviewer_239z · 2022-07-10

**Rating:** 6
**Confidence:** 4
**Soundness:** 3 good
**Presentation:** 2 fair
**Contribution:** 3 good

**Summary:**

The paper proposes a method to aggregate crowdsourcing answers based on a key observation: experts have different expertise levels and experts at a higher level are able to “simulate” the experts at lower levels. The paper aims to find this underlying “thinking hierarchy” —which can be used to find the best answer—by asking people to predict other people’s answers. The paper first proposes a model for this thinking hierarchy. The model assumes that experts have their types, and for each type, there is a thinking oracle that specifies how this type of experts generate their answers. An expert can run the oracles with lower types but never higher types. The paper then develops two algorithms to learn the underlying thinking hierarchy. The algorithms basically generate an answer-prediction matrix based on experts’ answers and predictions and then find a ranking of answers that will reorder the matrix to match the hierarchical structure in the best way. The paper provides theoretical justification for their algorithms. Finally, the paper shows by real-world experiments that their methods outperform plurality voting.


**Questions:**

1. Why not just assume there is an underlying ranking of answers and people with higher-type answers can predict the lower-type answers?

2. How do your algorithms compute the best permutation matrix/the matrix W*? Are they polynomial-time algorithms?


**Limitations:**

Yes.

**Strengths And Weaknesses:**

The strengths of the paper are the algorithms and the experiments. The paper proposes novel methods to utilize the underlying thinking hierarchy when aggregating experts’ answers and tests the method through real-world experiments. The algorithms are intuitively reasonable and the experiment results are good.

The weakness of the paper is the modeling of the thinking hierarchy, which also makes the theoretical justification of the algorithms implausible. The use of thinking oracle seems unreasonable to me. Are these thinking oracles private information or public information? If they are private information, how can the experts of higher types use the lower-type oracles? If they are public information, why can’t the lower-type experts use the higher-type oracles? It is also ungrounded why an expert generates predictions by running thinking oracles of others. The assumption seems crucial for the theoretical analysis, which in my point of view should be more carefully justified.

---

> ### Author Response · Authors · 2022-07-31
> **Modelling Justification &. Algorithm Efficiency**
>
> Thank you very much for your review!
>
> Q: Modeling Issue: are thinking oracles public or private? how to generate prediction?
>
> A: All of these oracles are private information. People can run some of them privately during the thinking process. Thus, it does not mean type t respondent can only run the oracle of her own type, t, because a type t agent may have run type < t oracle during her thinking process.
>
> Our model is inspired by two prior works:
>
> 1) System 1/system 2 theory in (Tversky and Kahneman. Judgment under uncertainty: Heuristics and biases. Science, 1974.).
>
> This theory assumes that people have two systems, a fast and intuitive system, and a slow and logical system. In our language, running the system is running the oracles.
>
> For example, when an expert starts to solve the circle problem, when she read the question, she will first run her intuitive system 1 and obtain answer 3. However, when she starts to think carefully, she runs her more careful system 2 to obtain answer 4. Using our language, she first runs a lower-type oracle and then runs a higher-type oracle.
>
> One of our contributions is to show that people who have run system 2 have also run system 1, but not vice versa.
>
> We extend the system 1/system 2 theory to the existence of multiple oracles because, during the thinking process, the respondent may approach the question through different thinking paths.
>
> 2) Cognitive hierarchy theory (CHT) (Camerer, Ho, and Chong. A cognitive hierarchy model of games. The Quarterly Journal of Economics, 2004.)
>
> This theory assumes that when agents play games, they perform different levels of strategies. Level k strategy is the best response to the < k level strategies.  The agent starts from the lowest level and then derives a higher level. For example, in the famous Beauty contest game（Camerer, Ho, and Chong 2004 ), people are asked to guess 2/3 of the average of people's answers, the theory assumes that a person starts from a random number as level 0 answer and then obtain 2/3 of the average of the level 0 answer, which is the level 1 answer, and the obtain 2/3 of the average of the mix of level 0&1 answers., which is the level 2 answer....
>
> In the above theory, the system and strategy are functions, which correspond to the oracles in our model. In the cognitive hierarchy theory, before people adopt the high-level strategy, they usually have run a low-level strategy. Our model has a similar conceptual idea. Though, unlike CHT, our approach does not need to understand the problem to obtain the thinking hierarchy.
>
> Q: Why an expert generates predictions by running thinking oracles of others
>
> A: When an agent generates predictions for other people's answers, she will output the "mistake" she has made before. Therefore, she will output an answer output by the oracles she has run before during the thinking process.
>
> Other Questions:
>
> Q: Why not just assume there is an underlying ranking of answers and people with higher-type answers can predict the lower-type answers?
>
> A: Our model is more general. We provide a more general model mainly for the following two reasons:
>
> a) Our model does not require the higher-type answer to be able to predict ALL lower types. In the above example, it is also possible that some higher-type respondents obtain the correct answer initially by running system 2 directly without running system 1. Thus, we make the assumption of upper-triangularity *without* assuming all entries in the upper-triangular part are non-zero, *nor* assuming all entries are one.
>
> b) Our model allows one type to output more than one answer. For example, there are two types. The higher type outputs 4, the lower type outputs 3 or 6. In this case, our model allows people who answer 3 and people who answer 6 can predict each other's answers. In this case, 3 & 6 will form a community and 4 is higher than any of it.
>
> We observe both a) and b) in the results collected by empirical studies.
>
> Q: How do your algorithms compute the best permutation matrix/the matrix W*? Are they polynomial-time algorithms?
>
> A: We use dynamic programming to compute the best permutation matrix, more details and pseudocode are in Section B. It's not a polynomial of the input size.
>
> However, the input size of the algorithm is the size of the answer space. In practice, we can set a  threshold t and only collect answers that are supported by more than t% of people. For example, in our empirical studies, we set the threshold as 3% (See line 274, Section 3 Studies). In this case, the number of distinct answers is at most 1/3% ~= 33, which is a constant. Therefore, the time complexity is a *constant* when the threshold t is a constant.
>
> In our empirical studies, the size of the answer space is at most 7 or 8. Our default algorithm takes only **91 milliseconds** to finish the computation of all 152 questions.
>
> Therefore, our algorithm's efficiency is sufficient for broad applications.

---

> > ### Comment · Reviewer_239z · 2022-08-08
> > **Questions addressed; Score raised to 6**
> >
> > The authors addressed my main concerns and questions about the model. I raise my score to 6. It would be nice to include these explanations about the model in the revised paper.

---

> > > ### Author Response · Authors · 2022-08-09
> > > **Thanks**
> > >
> > > Thanks for your reply! We will add these explanations about the model in the revised paper.

---

### Official Review · Reviewer_J65o · 2022-07-12

**Rating:** 5
**Confidence:** 3
**Soundness:** 3 good
**Presentation:** 3 good
**Contribution:** 3 good

**Summary:**

The paper proposes a mathematical model for thinking hierarchy of users predictions for their answer and other users’ answers. Given the joint distribution of the answers and predictions of other answers, the authors show that the parameters of the model can be derived using a novel matrix factorization. The authors solve an equivalent Frobenius norm minimization problem for the special case when the $W$ matrix (one of the parameters of the model) is semi-orthogonal. And then the authors propose a brute-force search based algorithm to find the ranking of the answers from the thinking hierarchy. Since the joint distribution of answers and predictions may not be available they use an empirical estimate. Naturally, this works if the samples are i.i.d.

**Questions:**

1. Lines 155 to 161 are confusingly written. Are the people predicting other peoples’ answers or their types?
2. Please write Example 2.3 properly. Since the meaning of $p_t$ and $p_{t \rightarrow t’}$ has already been explained it would be much easier to understand if the authors just use this notation while showing the construction of $\Lambda$.
3. Could the authors comment about what happens if the side information about other people’s predictions is noisy?

**Limitations:**

I don’t see any mention of the negative impacts of when the proposed algorithms are bad compared to standard algorithms. This discussion is needed.

**Strengths And Weaknesses:**

Strengths:

1. The problem of eliciting thinking hierarchy to come up with the correct ranking over various answers is well-motivated, and is an important problem to study.
2. The paper seems to be placed well in the recent literature about using additional information about people’s prediction about other answers to get better accuracy for ranking answers.
3. The theoretical results about the problem are interesting, however somewhat limited.

Weaknesses:

1. The main drawback is that the algorithms proposed in the paper are just brute-force search based algorithms and hence are not efficient for complicated questions.
2. The theoretical guarantees about eliciting thinking hierarchies hold in fairly restricted settings, which may not hold majority of the times in real-world. Especially, the questions and answers very quickly become complicated, in which case, plurality-voting could be a better method?
3. Basically, for a complete picture, the complexity of the questions and answers needs to be captured in this framework. In what cases does plurality-voting give better ranking than the algorithms in the proposed framework. I don’t understand when is the side information better?
4. The writing can be improved (see questions).

---

> ### Author Response · Authors · 2022-07-31
> **Efficiency of the algorithm & Complexity of Questions & Robustness & Theoretical Assumptions**
>
> Thank you very much for your review!
>
> Q: Efficiency of Algorithm.
>
> A: The input size of the algorithm is the size of the answer space. In practice, we can set a  threshold t and only collect answers that are supported by more than t% of people to avoid noise. For example, in our empirical studies, we set t=3% (See line 274, Section 3). Here the number of distinct answers is at most 1/3% ~= 33, which is a constant. Therefore, the time complexity is a constant when the threshold t is a constant.
>
> We also use dynamic programming to improve efficiency (see line 463, Section B). In our empirical studies, the size of the answer space is at most 7 or 8. Our default algorithm takes only **91 milliseconds** to finish the computation of all 152 questions.
>
> Therefore, our algorithm's efficiency is sufficient for broad applications.
>
> Q: Restricted settings:
>
> A: Without the thinking hierarchy assumption, we cannot deal with the systematic bias of the majority like in the scenario of the circle problem. This assumption is conceptually very similar to the cognitive hierarchy assumption (Camerer, Ho, and Chong 2004. ) (see more comments for the model in reply to Reviewer 239z ).
>
> In the empirical studies, the lack of fit score and results illustrated that the answer-prediction matrix has a hierarchical structure as our theory predicted (See Figure 3(b), Section 3) in many situations. This validates thinking hierarchy assumption in various situations.
>
> Q: When is the method better compared to the plurality vote?
>
> A: Theoretically, with thinking hierarchy assumptions, the top ranking answer of our method has the highest level, thus >= the level of plurality answer. Empirically, the studies illustrated the superiority of our method. In detail, possible scenarios include:
>
> 1)  Systematic bias exists: a) most people have a bias b) a few have the bias
> 2)  Mistakes are diverse: a) people do not know what others will answer, and the answer-prediction matrix is diagonal; b) people can easily predict each other's answer and the answer-prediction matrix does not have a hierarchical structure
>
> Our method is strictly better than the plurality in 1) a);  the same as the plurality in 1) b) and 2) a).
>
> In 2) b), plurality may be better because there does not exist a hierarchical score and our approach may output a random answer. However, in this case, the answer-prediction matrix has a high lack-of-fit score (see Definition 2.6 and figure3(b)), which we can judge at the beginning without any prior and choose to use a plurality vote.
>
> Q: What if the questions & answers are complex?
>
> A: We are not sure about the definition of complexity and provide comments based on our understanding.
>
> 1) The size of the answer space: with a threshold of 3%, the number of distinct answers is at most = 33 (see comments for efficiency of the algorithm). Moreover,  in many crowdsourcing applications including species recognition (Silvertown, et al 2015), the number of distinct answers can be even less like our empirical studies.
>
> 2) The format of the answers: for example, the answers are sentences. In this case, it's also not clear how to use the plurality-voting without classifying the answers. If the answers can be classified, then our method can be applicable again.
>
> 3) Difficulty: one contribution of our paper is a definition for the difficult questions: the question that most people make systematic bias and the answer-prediction matrix has a significant hierarchical structure. In this case, the plurality answer has a lower level and our approach is better than the plurality vote.
>
> Thus, our method can be implemented directly in many applications.
>
> Other questions:
>
> Q:  Lines 155 to 161...
> A: People do not need to predict types. It means "runs type t' oracle and report its output as a prediction". We will clarify it.
>
> Q: Please write ...
> A: We will follow your suggestion and clarify it.
>
> Q: Noisy side information about other people’s predictions?
> A: Here are our methods for robustness
> 1) Setting a threshold t% (see line 274, Section 3):
>
> The answer space A is the set of answers that are supported by more than t% of the respondents. The answer-prediction matrix only counts the number of people who answer in A and predicts an answer in A. For example, if a person predicts "Bulbasaur", but  "Bulbasaur" is not supported by t% of the population. This prediction will not be counted.
>
> 2) Finding the closest solution:
>
> We find the closest solution and use the Frobenius norm to measure the distance (see definition for NCT). This is similar to low-rank approximation and will be robust to data noise when agents' behavior does not fully follow our model. For example, in the circle problem, a few people who answer low-level answer "3" predicts high-level answer "4" in the collected data. However, because our algorithm finds the rank that maximizes the sum of the square in the upper-triangular area, the result still ranks "4" as the highest level.

---

> > ### Author Response · Authors · 2022-08-09
> > **Comments addressed?**
> >
> > Does our reply address your comments?  We wondered if you had any additional questions for us. Thanks!

---

> > > ### Comment · Reviewer_J65o · 2022-08-09
> > > **running time**
> > >
> > > Could the authors expand further on the theoretical running time bound for the dynamic programming algorithm?
> > >
> > > Thanks for addressing all other comments.

---

> > > > ### Author Response · Authors · 2022-08-09
> > > > **Theoretic running time**
> > > >
> > > > Thanks for your reply!
> > > >
> > > > The input size is the size of the answer space |A|. The brute force requires $O(|A|! * |A|^2)$. The dynamic programming algorithm improves the complexity to $O(2^{|A|}*|A|^2)$ (see lines 460-467 in the full version attached in the supplementary material).
> > > >
> > > > We want to additionally emphasize that
> > > >
> > > > 1) the input size  |A|<=33 when we set threshold t = 3% and our algorithm is very efficient empirically as we mentioned in our previous comment.
> > > >
> > > >
> > > > 2) The main focus of this paper is not proposing a more efficient algorithm to improve an existing algorithm that is notorious for slow running time previously. The main focus of this paper is a novel model and approach to address the systematic bias in crowdsourcing. For practical use, the algorithm is sufficiently efficient and outputs an accurate answer without any prior even if the majority is wrong.
> > > >
> > > >
> > > > Hope our reply can address your comments! Thank you very much!

---

> > > > > ### Comment · Reviewer_J65o · 2022-08-10
> > > > > **final response**
> > > > >
> > > > > I suggest the authors put the running time analysis in the main body of the paper, probably even the dynamic programming-based algorithm.
> > > > >
> > > > > I think the exponential running time in the number of possible answers is not very practical and limits the application of the model proposed in the paper with the algorithms known so far. However, the authors clarify, and I agree, that their main contribution is the mathematical model of the thinking hierarchy. Hence, raising my score to 5.

---

### Official Review · Reviewer_SfLf · 2022-07-19

**Rating:** 7
**Confidence:** 1
**Soundness:** 3 good
**Presentation:** 2 fair
**Contribution:** 3 good

**Summary:**

The crux of this paper is that it provides an empirically validated way to use the wisdom of the crowd rather than the default “popular answer”/”plurality voting” paradigm. In the process, the authors show how to obtain the thinking hierarchy of the crowd for the given set of questions. They claim that knowing this (rich) hierarchy helps in areas like policy making. Using mathematical tools and certain assumptions, the paper demonstrates the superiority of their method especially when obtaining a prior distribution from the crowd is prohibitively expensive.


**Questions:**

See above.

**Limitations:**

The authors discuss technical limitations such as the i.i.d assumption on user predictions. Negative impacts on society are not particularly discussed, but one could think of situations such as political mercenaries or governments collecting information on thinking patterns of a voting population in pursuit of an agenda.


**Strengths And Weaknesses:**

Originality: The authors distinguish themselves from previous work by discarding the use of a prior distribution. Further, they claim that their thinking hierarchy learning model captures a richer spectrum of answers. Their method reduces bias like previous work does, with the difference that they do not collect a prediction distribution, rather a single prediction answer. They also differentiate themselves from the game theoretic setting. The concept of the thinking hierarchy is sufficiently novel wrt past work.

Quality: The paper provides mathematical justification, although under certain idealistic assumptions, for every statement it makes. The experiment results have been provided at a URL and are easy to grasp. The authors are upfront about the assumptions that are required and also provide alternatives - for e.g. the i.i.d assumption under which respondents make their predictions is contrasted with picking the first prediction that a respondent makes, due to the fact that i.i.d predictions don’t match with reality.

Clarity: The paper is clear with what it aims to achieve and under what conditions it can achieve them. The goal is to: learn the thinking hierarchy among respondents and to do so without collecting prior information. The authors also provide future uses of their work from the ML and scale points of view. The experiments are easy enough for a novice reader to understand, and collect sufficient information.

Significance: The method described in the paper has significance in that it is applicable to cases where collecting prior information from a crowd is difficult, and plurality voting is not sufficient. One can foresee future applications of systematically obtaining a thought hierarchy as more information on a certain topic or policy can only help with decision making.

---

> ### Author Response · Authors · 2022-07-31
> **Potential Negative Impact**
>
> Thank you very much for your review!
>
> Question: Negative impacts on society are not particularly discussed, but one could think of situations such as political mercenaries or governments collecting information on the thinking patterns of a voting population in pursuit of an agenda.
>
> Answer: Thanks for mentioning this. We appreciate your recognition of the potential of our approach. Yes, our algorithm can be used by governments or political actors to elicit the hierarchy of opinions from the crowds. With the thinking pattern information, it may be easier to implement a social media manipulation of public opinion. We will add more discussions on the negative impacts of our approach in the revised version.

---

### Official Review · Reviewer_93Te · 2022-07-27

**Rating:** 6
**Confidence:** 3
**Soundness:** 3 good
**Presentation:** 3 good
**Contribution:** 3 good

**Summary:**

This paper argues that in the wisdom-of-the-crowd paradigm, plurality voting may not necessarily yield the correct answer when the majority makes systematic errors. The paper presents a theoretical framework to elicit thinking hierarchy and demonstrates that their method outperforms plurality voting and also demonstrates certain desirable properties. Apart from presenting the theoretical framework, the paper also conducts crowdsourced user studies to demonstrate the practical effectiveness of their framework.

**Questions:**

1. How are the questions for the user study selected?

**Limitations:**

No negative societal impact is discussed. I do not see any obvious red flags in terms of negative societal impact.

**Strengths And Weaknesses:**

Strengths:
1. The paper's motivation is clear and important. If a vast majority displays certain biases, plurality voting would not be very useful.
2. The paper's writing and organization are good. The examples provided help understanding.

Weaknesses:
1. The paper can be strengthened by making the user study more elaborate with a clear description of how are the questions selected.

---

> ### Author Response · Authors · 2022-07-31
> **Selecting Questions in User Study**
>
> Thank you very much for your review!
>
> Question: How are the questions for the user study selected?
>
> Answer: Except some questions in math study are classic interview problems like the Monty Hall problem, the questions are selected randomly from a pool. For example, the Life-and-death problems are selected from the quiz site (https://www.101weiqi.com/). The math problems are selected from a question bank which consists of math Olympiad contest problems for elementary school.
>
> We talked about it in Section A of our full version paper (the full version is in the supplementary materials) and will add more discussions.

---

### Meta-Review · Area_Chair_199j · 2022-08-23

**Recommendation:** Accept
**Confidence:** Less certain

**Metareview:**

This work proposes the framework to elicit people's "thinking hierarchy" that helps improve the wisdom of the crowd even if the majority is wrong. The reviewers overall appreciate the main idea of the work and believe it makes a nice contribution to the literature.  There have been some questions/concerns raised about the efficiency of the algorithm and the model limitations, to which the authors have provided reasonable responses.  We encourage the authors to incorporate those responses (and other reviewer comments) into the final version of the paper.

**Award:**

No

---

### Decision · Program_Chairs · 2022-09-14

Accept